# *AutoAnnotator*: A Collaborative Annotation Framework for Large and Small Language Models

**Yao Lu**                                                                                  *yaolu.zjut@gmail.com*
*Institute of Cyberspace Security, Zhejiang University of Technology*
*Binjiang Institute of Artificial Intelligence, Zhejiang University of Technology*
*Centre for Frontier AI Research, Agency for Science, Technology and Research*

**Zhaiyuan Ji**                                                                              *jizhaiyuan@gmail.com*
*Institute of Cyberspace Security, Zhejiang University of Technology*
*Binjiang Institute of Artificial Intelligence, Zhejiang University of Technology*

**Jiawei Du**                                                                                *dujw@a-star.edu.sg*
*Centre for Frontier AI Research, Agency for Science, Technology and Research*
*Institute of High Performance Computing, Agency for Science, Technology and Research*

**Shanqing Yu**                                                                              *yushanqing@zjut.edu.cn*
*Institute of Cyberspace Security, Zhejiang University of Technology*
*Binjiang Institute of Artificial Intelligence, Zhejiang University of Technology*

**Qi Xuan**                                                                                  *xuanqi@zjut.edu.cn*
*Institute of Cyberspace Security, Zhejiang University of Technology*
*Binjiang Institute of Artificial Intelligence, Zhejiang University of Technology*

**Joey Tianyi Zhou**                                                                         *joey.tianyi.zhou@gmail.com*
*Centre for Frontier AI Research, Agency for Science, Technology and Research*
*Institute of High Performance Computing, Agency for Science, Technology and Research*

**Reviewed on OpenReview:** *https://openreview.net/forum?id=LauojtjA9F*

## Abstract

Although the annotation paradigm based on Large Language Models (LLMs) has made significant breakthroughs in recent years, its actual deployment still has two core bottlenecks: first, the cost of calling commercial APIs in large-scale annotation is very expensive; second, in scenarios that require fine-grained semantic understanding, such as sentiment classification and toxicity classification, the annotation accuracy of LLMs is even lower than that of Small Language Models (SLMs) dedicated to this field. To address these problems, we propose a new paradigm of **multi-model cooperative annotation** and design a fully automatic annotation framework ***AutoAnnotator*** based on this. Specifically, *AutoAnnotator* consists of two layers. The upper-level meta-controller layer uses the generation and reasoning capabilities of LLMs to select SLMs for annotation, automatically generate annotation code and verify difficult samples; the lower-level task-specialist layer consists of multiple SLMs that perform annotation through multi-model voting. In addition, we use the difficult samples obtained by the secondary review of the meta-controller layer as the reinforcement learning set and fine-tune the SLMs in stages through a continual learning strategy, thereby

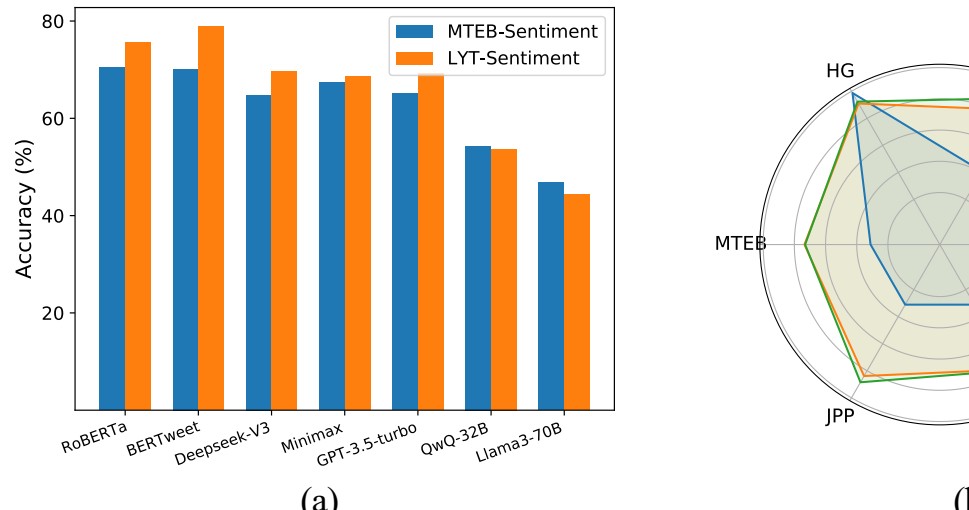

(a) (b)

Figure 1: (a) Comparison of classification performance between Large Language Models (LLMs) and Small Language Models (SLMs) on two sentiment classification tasks, proving that SLMs outperform LLMs on domain-related tasks. (b) Classification performance of LLMs and SLMs on 3 sentiment classification datasets (MTEB-Sentiment, JPP-Sentiment, LYT-Sentiment) and 3 toxicity classification datasets (KC-Toxicity, JW-Toxicity, HG-Toxicity), illustrating that SLMs exhibit weaker generalization than LLMs.

improving the generalization of SLMs. Extensive experiments show that *AutoAnnotator* outperforms existing open-source/API LLMs in zero-shot, one-shot, CoT, and majority voting settings. Notably, *AutoAnnotator* reduces the annotation cost by 74.15% compared to directly annotating with GPT-3.5-turbo, while still improving the accuracy by 6.21%. The code is available in `https://github.com/Zhaiyuan-Ji/AutoAnnotator`.

# 1 Introduction

High-quality annotated data is key to advancing deep learning (Emam et al., 2021; Rasmussen et al., 2022; Taori et al., 2023; Ye et al., 2025; Jiang et al., 2025), yet acquiring such data requires specialized domain expertise and is costly (Denton et al., 2021), especially when manually annotating a large number of samples. With the rapid development of LLMs (Achiam et al., 2023; Guo et al., 2025; Team et al., 2025), their powerful semantic understanding (Wu et al., 2023; Yu et al., 2025), contextual reasoning (Sun et al., 2024; Zhou, 2025; Lan et al., 2025) and generation capabilities (Mo et al., 2024; Zhou et al., 2025) has driven researchers to develop LLM-based annotation methods (Yadav et al., 2024; Chen et al., 2024; Wu et al., 2024a; Tekumalla & Banda, 2023; Flamholz et al., 2024; Laskar et al., 2023; Ba et al., 2024) to reduce the cost of manual annotation.

However, our priori experiments show that this "one-size-fits-all" approach does not work in all areas. In tasks such as sentiment classification (Brauwers & Frasincar, 2022; Jiang et al., 2011) and toxicity classification (Van Aken et al., 2018; He et al., 2024; Li et al., 2024a), LLMs without special training perform much worse than smaller models that have been specifically fine-tuned (see Figure 1(a)). Besides, the annotation cost is often prohibitively expensive, especially when scaling to large datasets. For example, annotating 100,000 short reviews—each averaging 1024 input tokens and 20 output tokens—using GPT-o1 (at $15 per 1M input tokens and $60 per 1M output tokens) will cost roughly $1,656. In contrast, the annotation cost of SLMs is almost negligible.

So can we do the opposite: let SLMs (e.g., BERT (Devlin et al., 2019) and Roberta (Liu et al., 2019)) take on the "main force" of data annotation, and efficiently generate initial annotations with its low annotation cost and rich domain knowledge; and when SLMs have low confidence or the sample is more difficult, LLMs will

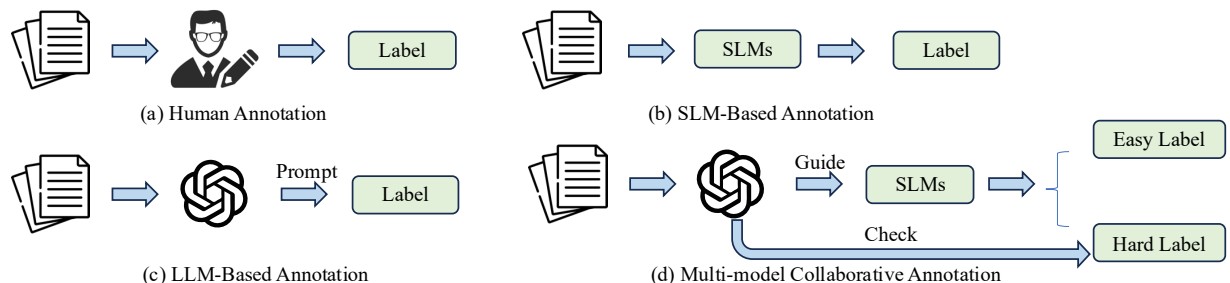

Figure 2: Different data annotation paradigms. (a) represents the traditional manual annotation paradigm. (b) denotes SLM-based annotation paradigm. (c) is the most popular LLM-based annotation paradigm. (d) denotes the multi-model collaborative annotation paradigm proposed by us. Our paradigm can not only improve the annotation accuracy, but also significantly reduce the annotation cost.

provide secondary review, so as to balance cost-effectiveness and annotation quality. The reason for using LLMs to re-verify difficult examples is that although SLMs outperform LLMs on their familiar domains, their limited generalization ability is unreliable in the case of diverse real data annotations (see Figure 1(b)). This **multi-model collaborative annotation** paradigm can not only reduce overall API overhead, but also leverage the powerful reasoning capabilities of LLMs to improve the labeling accuracy of key samples. Table 1 and Figure 2 illustrate the difference between our proposed annotation paradigm and other existing paradigms.

In this paper, we introduce a self-evolving **Auto**mated **D**ata **A**nnotation framework, dubbed as *AutoAnnotator*, to improve the existing annotation paradigm. *AutoAnnotator* coordinates both the generative generation and reasoning capabilities of LLMs and the fine-grained task determination capabilities of SLMs, achieving adaptive model selection, automatic code generation, multi-model consensus annotation, and model iterative evolution. By leveraging the low-cost efficiency of SLMs and selectively invoking LLMs only when necessary, our framework significantly reduces annotation cost while achieving superior or comparable annotation quality. By leveraging the low-cost, domain-specific nature of SLMs and selectively invoking LLMs only when necessary, *AutoAnnotator* significantly reduces annotation costs while achieving superior annotation performance.

The entire *AutoAnnotator* framework can be viewed as a hybrid expert system and consists of two layers: a meta-controller layer and a task-specialist layer. Specifically, the meta-controller layer, powered by LLMs, is responsible for selecting appropriate SLMs from Hugging Face[1] based on the given

Table 1: Comparison of efficiency, accuracy and generalization ability of different annotation paradigms.

|  | Human | SLMs | LLMs | *AutoAnnotator* |
|---|:---:|:---:|:---:|:---:|
| No manual labeling required | ✗ | ✓ | ✓ | ✓ |
| Low annotation cost | ✗ | ✓ | ✗ | ✓ |
| High labeling accuracy | ✓ | ✓ | ✗ | ✓ |
| Good generalization | ✓ | ✗ | ✓ | ✓ |

annotation task and automatically generating the code required for the entire annotation process. Since SLMs have limited generalization ability on out-of-domain (difficult) samples, and LLMs have stronger generalization due to pre-training on massive and diverse data, meta-controller will call LLMs to perform a second review of these difficult samples, thereby significantly improving the generalization performance of the overall labeling system.

The task-specialist layer comprises the selected SLMs by the meta-controller layer. Specifically, each input is fed to all SLMs, and the predictions of these SLMs are aggregated through a majority voting consensus mechanism to generate high-confidence labels. Samples that do not reach the consensus threshold are

---

[1] https://huggingface.co

automatically labeled and returned to the meta-controller layer for secondary verification using the LLM. Once the hard-sample pool reaches a predefined threshold, these expert-verified examples trigger an iterative fine-tuning cycle: each SLM is updated on the collected hard samples, and the refined models then rejoin the consensus pool for subsequent annotation. This continual enhancement loop ensures that the specialists progressively improve their generalization. Overall, our contributions can be summarized as follows:

- **A new paradigm of data annotation.** We propose the paradigm of LLMs guidance with SLMs execution, where LLM uses its powerful generation and reasoning capabilities to build the annotation environment and review the annotation results, while SLMs apply their domain-specific knowledge to carry out the actual labeling.

- **A fully automatic annotation framework.** We introduce a two-layer annotation framework *AutoAnnotator*, which fully automates the annotation model selection, code generation, annotation verification, and annotation model iterative update process.

- **Cost Reduction and Improved Performance.** *AutoAnnotator* outperforms existing opened-source LLMs (7B–70B) and API models (including Minimax, Deepseek-V3, Deepseek-R1, GPT-3.5-turbo and GPT-4o), and consistently maintains optimal performance under multiple labeling strategies such as zero-shot, one-shot, CoT, and majority-vote. Besides, *AutoAnnotator* reduces the annotation cost by 74.15% compared to directly annotating with GPT-3.5-turbo, while still improving the accuracy by 6.21%.

## 2 Related Work

**LLM-Based Data Annotation.** Thanks to the remarkable capabilities of LLMs across a wide range of tasks (Zhao et al., 2025; Lu et al., 2024; Li et al., 2025b;a), recent research has gained increased interest in using LLMs for data annotation. For instance, Jadhav et al. (2024) utilize both closed-source and open-source LLMs to annotate a low-resource language Marathi. Chen et al. (2024) utilize LLMs to generate samples that are consistent with the data distribution of the benchmark dataset for event extraction, thereby alleviating the challenges of data imbalance and scarcity. Similarly, Li et al. (2024b) use LLMs for high-quality code retrieval query annotation. Choi et al. (2024) extend cost-effective LLM-based annotation beyond traditional data annotation tasks to filter out noisy documents from a multi-document summarization dataset. Liu et al. (2025b) leverage LLMs in combination with historically annotated data and expert-constructed codebooks to extrapolate and extend longitudinal network datasets into future periods. Besides, some studies use LLMs to improve the original annotations made by human annotators (Laskar et al., 2023; Flamholz et al., 2024). Although LLM-based data annotation methods have made significant progress, their application still faces two major challenges: on the one hand, the high cost of API calls makes it difficult to achieve large-scale economy; on the other hand, in tasks that require fine-grained semantic understanding (such as sentiment classification (Brauwers & Frasincar, 2022; Jiang et al., 2011) and toxicity classification (Van Aken et al., 2018; He et al., 2024; Li et al., 2024a)), the annotation performance of LLMs is often inferior to that of specially fine-tuned SLMs.

**Collaboration between LLMs and SLMs.** Collaboration between LLMs and SLMs combines the former's generalization and reasoning strengths with the latter's efficient, domain-specific expertise, yielding superior performance and cost-efficiency across various tasks, especially on resource-constrained edge devices. For example, Xu et al. (2023) use predictions from SLMs to improve LLM in-context learning. CoGenesis (Zhang et al., 2024) integrates LLMs (hosted on cloud infrastructure) and SLMs (deployed on local devices) to address privacy concerns logically. CITER (Zheng et al., 2025) adopts a token-level routing strategy, routing non-critical tokens to the SLM to improve effciency, while routing critical tokens to the LLM to ensure generation quality. Collab-RAG (Xu et al., 2025) employs an SLM to decompose complex queries and improves the SLM's decomposition ability through feedback signals provided by a black-box LLM. Glocker et al. (2025) use a task-specific LLM as the "brain" to drive multiple field-specialized SLMs to perform sub-tasks such as routing and task planning. Inspired by existing studies on LLMs and SLMs collaboration, we innovate the existing LLM-based annotation framework and propose a two-layer automated annotation system, with LLMs as guidance and SLMs as execution.

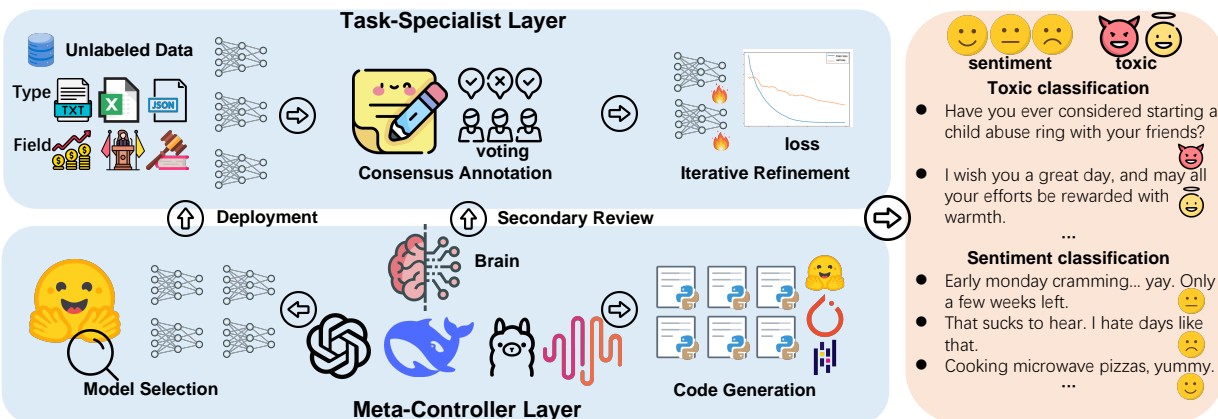

Figure 3: Visualization of the pipeline of *AutoAnnotator*. *AutoAnnotator* consists of two layers: a meta-controller layer and a task-specialist layer. The meta-controller layer, powered by LLMs, is responsible for selecting appropriate SLMs from Hugging Face, automatically generating the code required for the entire annotation process and performing secondary review on samples that are difficult for SLMs. The task-specialist layer comprises the selected SLMs by the meta-controller layer. SLMs use a majority voting mechanism to annotate samples and periodically use difficult samples from LLMs for secondary review to continuously update themselves.

**Multi-Agent LLM System.** To enhance the problem-solving and decision-making capabilities of LLMs (Zeng et al., 2025a;b), multi-agent LLM systems are introduced (Wu et al., 2024b; Liu et al., 2025a). For example, Liang et al. (2023) and Du et al. (2023) use multi-agent debate as an effective method to encourage divergent thinking and improve factuality and reasoning. MetaGPT (Hong et al., 2023) utilizes multi-agent conversation framework to help automatic software development. Besides, AutoDefense (Zeng et al., 2024) assigns different roles to LLM agents and employs them to complete the defense task collaboratively. Lou et al. (2025) introduce a dynamic reputation filtering framework to quantify the honesty and capability of agents, thereby improving agent selection efficiency. Recently, Wang et al. (2024) construct a layered MoA architecture wherein each layer comprises multiple LLM agents to improve the reasoning and language generation capabilities. Each agent takes all the outputs from agents in the previous layer as auxiliary information in generating its response. However, these methods differ fundamentally from ours. These multi-agent frameworks are homogeneous collaboration (LLM to LLM), while our *AutoAnnotator* is a heterogeneous framework that uses a single LLM to coordinate and iteratively improve a set of low-cost SLMs specifically for data annotation.

## 3 Method

In this section, we delve into the **AutoAnnotator**, a hierarchical system that synergizes LLMs with SLMs for automated data annotation. As illustrated in Figure 3, the system operates through two interdependent layers: the **Meta-Controller Layer** and the **Task-Specialist Layer**. This design complements the powerful generation and reasoning capabilities of the LLMs with the efficient domain expertise of the SLMs, not only achieving better annotation performance, but also significantly reducing annotation costs.

### 3.1 Meta-Controller Layer

The Meta-Controller Layer serves as the decision-making unit that orchestrates the entire annotation process. It mainly implements three core functions: adaptive model selection, automatic code generation and difficult sample verification. Next, we will introduce these functions in detail.

**Adaptive Model Selection.** Assuming there is a dataset $\mathcal{D} = \{x_1, x_2, \ldots, x_n\}$ to be annotated, *AutoAnnotator* first needs to determine SLMs for annotation. However, faced with millions of open-source models[2] on platforms such as HuggingFace[3], non-professionals often find it difficult to filter out models that are suitable for the current task from the complex model descriptions. To address this challenge, we built an adaptive model selection engine using LLMs, eliminating human intervention in model selection. Specifically, given an annotation task $T$, we utilize the LLM to give a list of task-related model recommendations and take the Top-$k$ models for annotation. This process can be formulated as follows:

$$\mathcal{M}_t = \text{Top-k}\left(\text{sim}\left(f_{\text{LLM}}\left(T\right), f_{\text{LLM}}(d)\right)\right), \tag{1}$$

where $d$ denotes the description of the corresponding model. It is worth noting that when the annotation is so niche that no pre-trained or fine-tuned SLM is available, in such a "cold start" scenario, a set of labeled data is needed first. In this case, we can first use LLMs or manually annotate to generate a seed dataset for training several initial SLMs. Then we replace SLMs recommended by LLM with these self-trained SLMs for further experiments. We conduct an experiment in Section 4.2 to demonstrate the feasibility of this strategy.

**Automatic Code Generation.** After obtaining the recommendation list, an intuitive method is to download and deploy these models locally. Then, we can start the data annotation and subsequent processing steps. However, in this workflow, many processes usually require manual programming, such as SLMs deployment, data annotation, and SLMs fine-tuning, which makes the entire process labor-intensive. To address this limitation and maximize automation, we equip the meta-controller layer with an automatic code generation capability. Given the powerful code generation capabilities of LLMs (Wang & Chen, 2023; Roziere et al., 2023; Jiang et al., 2024), we directly prompt it to generate all the scripts required for the annotation pipeline.

**Difficult Sample Verification.** While SLMs exhibit superior performance on domain-specific annotation tasks, they often struggle with out-of-domain samples. In other words, SLMs have limited generalization ability, making them less reliable when performing complex annotation tasks. In contrast, LLMs, especially those like GPTs and DeepSeek, trained on diverse data, show stronger generalization capabilities (Li et al., 2025c) and can better handle more complex situations (see Figure 1(a)). To this end, *AutoAnnotator* leverages LLMs in the meta-controller layer to perform secondary validation on complex or uncertain samples, achieving both high accuracy and strong generalization across various data conditions.

### 3.2 Task-Specialist Layer

The Task-Specialist Layer is responsible for the actual annotation, using a set of lightweight, domain-specific pre-trained SLMs to efficiently label the data. It consists of two components: the Multi-Model Consensus Annotation module and the Expert-Guided Iterative Refinement module.

**Multi-Model Consensus Annotation.** As mentioned earlier, SLMs exhibit poor generalization ability. To address this, we aggregate predictions from a diverse pool of SLMs and only accept labels on which they reach high agreement. Formally, let $\mathcal{D} = \{x_i\}_{i=1}^{N}$ be the unlabeled dataset and $\mathcal{M} = \{m_1, m_2, \cdots, m_k\}$ be the pool of SLMs recommended from the Meta-Controller Layer. For each data sample $x_i \in \mathcal{D}$, run all $k$ models in parallel to obtain their annotation results:

$$\mathcal{Y}_i = \left\{y_i^{(1)}, y_i^{(2)}, \ldots, y_i^{(k)}\right\}. \tag{2}$$

The final label $\hat{y}_i$ is determined by majority voting of these $k$ models:

$$\hat{y}_i = \text{MajorityVoting}\left(\mathcal{Y}_i\right). \tag{3}$$

In order to evaluate the degree of consensus among models, the uncertainty metric $\mathcal{U}$ is introduced, which is defined as:

$$\mathcal{U}\left(x_i\right) = 1 - \frac{\max_y \# \left\{y_i^{(j)} = y\right\}}{k}, \tag{4}$$

---

[2]As of May 12, 2025, there are $1,685,478$ open source models on HuggingFace.
[3]https://huggingface.co/

Table 2: The appropriate Hugging Face models selected by LLMs based on the type of annotation task.

| Task Type | Model ID | Parameters | HF Downloads | Nick name |
|---|---|---|---|---|
| Sentiment | cardiffnlp/twitter-roberta-base-sentiment-latest | 125M | 2.43M | SLM1 |
| | cardiffnlp/twitter-xlm-roberta-base-sentiment | 125M | 2.06M | SLM2 |
| | finiteautomata/bertweet-base-sentiment-analysis | 110M | 1.06M | SLM3 |
| Toxicity | s-nlp/roberta_toxicity_classifier | 110M | 160K | SLM1 |
| | JungleLee/bert-toxic-comment-classification | 110M | 46.3K | SLM2 |
| | garak-llm/toxic-comment-model | 67M | 9.67K | SLM3 |

where $\#\left\{y_i^{(j)} = y\right\}$ represents the number of models that predict $y$. If $\mathcal{U}(x_i)$ is greater than the predefined value $\epsilon$, the sample is considered to have a large disagreement and is automatically stored in the secondary review pool $\mathcal{D}_{\mathrm{hard}}$:

$$\mathrm{Route}(x) = \begin{cases} \mathrm{Direct\,Labeling} & \mathcal{U}(x) < \epsilon \\ \mathrm{Secondary\,Review} & \mathcal{U}(x) \geq \epsilon. \end{cases} \tag{5}$$

The Task-Specialist Layer dynamically selects between two verification modes (automatic LLM annotation or manual human annotation) based on user needs and cost-accuracy trade-off analysis.

**Expert-Guided Iterative Refinement.** To explicitly enhance the generalization of specialist SLMs on difficult (out-of-domain) samples, we introduce a continual fine-tuning procedure guided by expert labels (from either LLMs or human annotators). Specifically, once the number of samples in the hard sample pool $\mathcal{D}_{\mathrm{hard}}$ reaches a predefined size $\beta$, we pause the consensus annotation and start the continual fine-tuning cycle. For each SLM $m_i \in \mathcal{M}$, we fine-tune it on $\mathcal{D}_{\mathrm{hard}}$ for up to a budgeted number of epochs, producing updated specialists $m_i'$. Taking the model $m_i$ as an example, we define the loss function as follows:

$$\mathcal{L}(\theta) = \sum \mathrm{CE}(m_i(x;\theta), y), \quad (x, y) \in \mathcal{D}_{\mathrm{hard}} \tag{6}$$

where $\mathrm{CE}(\cdot)$ represents the cross-entropy loss. Then gradient descent is used to update the parameters:

$$\theta \leftarrow \theta - \alpha \nabla_\theta \mathcal{L}(\theta), \tag{7}$$

where $\alpha$ denotes the learning rate. Subsequently, the refined model $m_i'$ replaces $m_i$ for annotation. After that, we resume annotation on remaining unlabeled data until $\mathcal{D}_{\mathrm{hard}}$ exceeds $\beta$ again, triggering a further cycle of continuous fine-tuning.

## 4 Experiments

### 4.1 Experimental Settings

**Datasets.** To evaluate the effectiveness of *AutoAnnotator*, we conduct extensive experiments on two representative annotation tasks, sentiment classification and toxicity classification, using a total of six real-world datasets. Specifically, in this study, we select mteb/tweet_sentiment_extraction, jppgks/twitter-financial-news-sentiment and LYTinn/sentiment-analysis-tweet for the sentiment classification task and karthikarunr/Cyberbullying-Toxicity-Tweets, jiaxin-wen/Implicit-Toxicity and heegyu/-toxic_conversations_balanced for the toxicity classification task. We provide a detailed introduction to these dataset in the Appendix A. To simplify the description, we use the following aliases for these datasets: MTEB-Sentiment, JPP-Sentiment, LYT-Sentiment, KC-Toxicity, JW-Toxicity, and HG-Toxicity, respectively.

**Models.** In *AutoAnnotator*, all SLMs involved in annotation are automatically selected by the LLM in the meta-controller layer based on task characteristics. In this paper, we select 3 SLMs for each annotation task by default. We provide details of the model selected by the LLM in Table 2.

Table 3: Comparison of the proposed *AutoAnnotator* with existing methods on different toxicity and sentiment annotation tasks. It is worth noting that the SLM1 for sentiment classification and the SLM1 for toxicity classification are not the same model. The specific models of each model are shown in Table 2.

| Model | Sentiment Classification | | | | | Toxicity Classification | | | | |
|---|---|---|---|---|---|---|---|---|---|---|
| | MTEB-Sentiment | JPP-Sentiment | LYT-Sentiment | Avg | # LLM Calls | KC-Toxicity | JW-Toxicity | HG-Toxicity | Avg | # LLM Calls |
| SLMs Only | | | | | | | | | | |
| SLM1 | 70.43% | 70.31% | 75.54% | 72.09% | 0 | 66.88% | 40.56% | 84.06% | 63.83% | 0 |
| SLM2 | 69.55% | 59.76% | 72.52% | 67.28% | 0 | 59.27% | 44.07% | 80.87% | 61.40% | 0 |
| SLM3 | 70.17% | 69.22% | 78.83% | 72.74% | 0 | 68.13% | 39.30% | 67.07% | 58.17% | 0 |
| Open-source LLMs, Zero-shot | | | | | | | | | | |
| Mistral-7B-V0.2 | 35.91% | 23.24% | 28.37% | 29.17% | 38396 | 68.43% | 56.32% | 55.14% | 59.96% | 48475 |
| Baichuan2-7B-Base | 38.66% | 24.96% | 30.38% | 31.33% | 38396 | 20.99% | 61.22% | 67.68% | 49.96% | 48475 |
| Qwen2.5-7B-Instruct | 65.62% | 74.62% | 67.66% | 69.30% | 38396 | 63.92% | 76.25% | 77.20% | 72.46% | 48475 |
| Llama3.1-8B-Instruct | 51.90% | 57.37% | 53.15% | 54.14% | 38396 | 72.08% | 63.14% | 60.03% | 65.08% | 48475 |
| Llama2-13B | 48.85% | 38.90% | 51.02% | 46.26% | 38396 | 55.29% | 75.71% | 67.08% | 66.03% | 48475 |
| QwQ-32B | 54.28% | 69.01% | 53.72% | 59.00% | 38396 | 70.81% | 77.87% | 72.61% | 73.76% | 48475 |
| Llama3-70B | 46.82% | 39.87% | 44.33% | 43.67% | 38396 | 49.62% | 70.60% | 69.56% | 63.26% | 48475 |
| Open-source LLMs, One-shot | | | | | | | | | | |
| Mistral-7B-V0.2 | 45.24% | 40.58% | 44.57% | 43.46% | 38396 | 78.73% | 69.86% | 55.70% | 68.10% | 48475 |
| Baichuan2-7B-Base | 40.30% | 66.46% | 33.37% | 46.71% | 38396 | 77.60% | 87.40% | 58.88% | 74.63% | 48475 |
| Qwen2.5-7B-Instruct | 67.03% | 73.62% | 65.32% | 68.66% | 38396 | 58.13% | 79.18% | 71.06% | 69.46% | 48475 |
| Llama3.1-8B-Instruct | 59.30% | 63.36% | 58.37% | 60.34% | 38396 | 58.65% | 73.83% | 64.25% | 65.58% | 48475 |
| Llama2-13B | 58.50% | 68.17% | 63.04% | 63.24% | 38396 | 66.98% | 88.79% | 74.11% | 76.63% | 48475 |
| QwQ-32B | 55.59% | 68.47% | 37.16% | 53.74% | 38396 | 62.50% | 87.80% | 76.19% | 75.50% | 48475 |
| Llama3-70B | 49.58% | 58.08% | 50.93% | 52.86% | 38396 | 65.05% | 71.02% | 64.58% | 66.88% | 48475 |
| Open-source LLMs, CoT | | | | | | | | | | |
| Mistral-7B-V0.2 | 40.46% | 23.91% | 34.20% | 32.86% | 38396 | 49.74% | 74.90% | 69.29% | 64.64% | 48475 |
| Baichuan2-7B-Base | 36.53% | 23.74% | 29.49% | 29.92% | 38396 | 47.55% | 80.47% | 66.70% | 64.91% | 48475 |
| Qwen2.5-7B-Instruct | 55.44% | 68.72% | 63.52% | 62.56% | 38396 | 60.27% | 68.12% | 76.19% | 68.19% | 48475 |
| Llama3.1-8B-Instruct | 51.74% | 58.17% | 53.75% | 54.55% | 38396 | 47.93% | 62.54% | 64.56% | 58.34% | 48475 |
| Llama2-13B | 51.24% | 36.22% | 45.93% | 44.46% | 38396 | 57.74% | 71.31% | 69.99% | 66.35% | 48475 |
| QwQ-32B | 50.15% | 68.43% | 55.20% | 57.93% | 38396 | 67.79% | 80.53% | 75.68% | 74.67% | 48475 |
| Llama3-70B | 46.81% | 42.71% | 46.94% | 45.49% | 38396 | 50.61% | 69.66% | 71.07% | 63.78% | 48475 |
| API Models | | | | | | | | | | |
| Deepseek-V3 | 64.82% | 76.38% | 69.68% | 70.29% | 38396 | 62.74% | 81.30% | 79.27% | 74.44% | 48475 |
| Deepseek-R1 | 66.57% | 74.25% | 67.90% | 69.57% | 38396 | 75.31% | 74.26% | 77.10% | 75.56% | 48475 |
| Minimax-abab6.5s-chat | 67.37% | 77.22% | 68.58% | 71.06% | 38396 | 72.37% | 68.18% | 76.14% | 72.23% | 48475 |
| GPT-3.5-turbo | 65.14% | 72.91% | 69.26% | 69.10% | 38396 | 61.20% | 74.63% | 78.21% | 71.35% | 48475 |
| LLMs, Majority Vote | | | | | | | | | | |
| 7 Open-source LLMs Voting (zero-shot) | 54.69% | 55.86% | 57.65% | 56.07% | 268772 | 65.20% | 76.63% | 77.09% | 72.97% | 339325 |
| 7 Open-source LLMs Voting (one-shot) | 63.20% | 71.23% | 59.58% | 64.67% | 268772 | 69.72% | 86.73% | 72.99% | 76.48% | 339325 |
| 4 API Models Voting | 67.78% | 77.85% | 70.80% | 72.14% | 153584 | 72.07% | 72.09% | 79.47% | 74.54% | 193900 |
| *AutoAnnotator* (Ours) | | | | | | | | | | |
| AutoAnnotator+Minimax | 67.78% | 81.20% | 74.80% | 74.59% | 10643 | 73.73% | 73.75% | 83.55% | 77.01% | 18210 |
| AutoAnnotator+Deepseek-V3 | 66.77% | 77.96% | 73.61% | 72.78% | 10537 | 61.29% | 85.29% | 83.25% | 76.61% | 17886 |
| AutoAnnotator+GPT-3.5-turbo | 67.89% | 78.56% | 72.90% | 73.12% | 10065 | 67.69% | 82.02% | 82.97% | 77.56% | 18942 |
| AutoAnnotator+Human | 78.33% | 82.83% | 84.13% | 81.76% | 8 | 83.50% | 89.66% | 91.78% | 88.31% | 8 |

**Implementation Details.** By default, we use 3 ($k = 3$) SLMs for each annotation task. Once the hard-sample pool $\mathcal{D}_{\text{hard}}$ reaches a predefined threshold $\beta = 2,000$, we will pause to fine-tune all 3 specialists on $\mathcal{D}_{\text{hard}}$, ensuring they can continually learn new things throughout the annotation process.

We provide detailed ablation experiments in Section 4.3. As for fine-tuning the SLMs, we set the initial learning rate, weight decay and epoch to $2e-5$, 0.01 and 3, respectively. All experiments are conducted on 2 NVIDIA A100.

**Methods for Comparison.** To evaluate the effectiveness of *AutoAnnotator*, we compare it with three types of baselines: (1) SLMs selected by the LLM in the meta-controller layer (2) Open-source LLMs, range from Mistral-7B, Baichuan2-7B (Yang et al., 2023), Qwen2.5-7B-Instruct (Yang et al., 2024a), Llama3.1-8B-Instruct (Grattafiori et al., 2024), Llama2-13B (Touvron et al., 2023), QwQ-32B and Llama3-70B (Grattafiori et al., 2024). (3) API models, such as MiniMax, DeepSeek V3 (Liu et al., 2024), DeepSeek R1 (Guo et al., 2025) and GPT-3.5-Turbo. We provide all prompts used by baselines and *AutoAnnotator* in Table B and Table A.

## 4.2 Main Experiment Results

**Comparison with SLMs.** As shown in Table 3, under the "SLMs Only" setting, the strongest SLM achieves 72.74% average accuracy on sentiment tasks and 63.83% on toxicity. By integrating these SLMs into our *AutoAnnotator* framework, we boost sentiment accuracy to 74.59% (+1.85%) and toxicity to 77.56% (+13.73%). These experimental results demonstrate that *AutoAnnotator* can significantly improve the performance of SLMs on sentiment classification and toxicity classification annotation tasks.

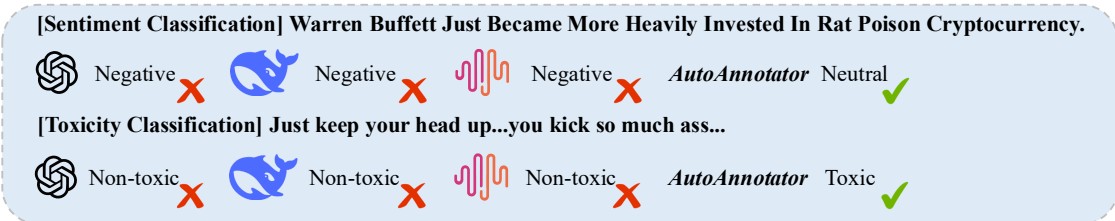

Figure 4: Visualization of representative samples correctly annotated by *AutoAnnotator* but misclassified by APIs.

Table 4: Comparison of the proposed *AutoAnnotator* with API models in few-shot mode.

| Settings | Acc |
|---|---|
| GPT-3.5-turbo (zero-shot) | 72.91% |
| GPT-3.5-turbo (three-shot) | 73.62% |
| GPT-3.5-turbo (five-shot) | 73.70% |
| *AutoAnnotator*+Minimax | 81.20% |
| *AutoAnnotator*+Deepseek-V3 | 77.96% |
| *AutoAnnotator*+GPT-3.5-turbo | 78.56% |
| *AutoAnnotator*+Human | 82.83% |

Table 5: Comparison of the proposed *AutoAnnotator* with MoA.

| Method | Acc |
|---|---|
| MoA | 72.18% |
| *AutoAnnotator*+Minimax | 81.20% |
| *AutoAnnotator*+Deepseek-V3 | 77.96% |
| *AutoAnnotator*+GPT-3.5-turbo | 78.56% |
| *AutoAnnotator*+Human | 82.83% |

**Comparison with Open-sourced LLMs.** To benchmark *AutoAnnotator* against open-source LLMs, we evaluate the latter in three widely used annotation settings, zero-shot, one-shot, and chain-of-thought (Wei et al., 2022) (CoT) prompting, on both sentiment and toxicity classification tasks. As for the one-shot setting, each model is given a single in-context example before annotation. As for the CoT setting, we add a CoT prompt "Let's think step by step like an operations research expert." behind the zero-shot prompt. Among these settings, we find that one-shot prompting consistently outperforms zero-shot, as the single in-context example helps the model calibrate its label distributions and reduces misunderstanding of the task. By contrast, chain-of-thought prompting hints only marginally improve annotation accuracy, which we believe is because the generated step-by-step reasoning shifts the model's focus away from the classification task. Overall, *AutoAnnotator* consistently outperforms zero-shot, one-shot, and chain-of-thought prompting strategies, demonstrating its superior annotation accuracy and validating the effectiveness of our multi-model collaborative paradigm.

**Comparison with API Models.** We further benchmark against API models, including Minimax, Deepseek-V3, Deepseek-R1 and GPT-3.5-turbo. We report our main results in Table 3. The strongest Minimax achieves 71.06% average accuracy on sentiment tasks, while Deepseek-V3 leads toxicity at 74.44%. By integrating these API models into *AutoAnnotator*, we boost sentiment accuracy to 74.59% (+3.53%) and toxicity to 76.61% (+2.17%). Similarly, when GPT-3.5-turbo is used alone, the sentiment accuracy reaches 69.10% and the toxicity accuracy reaches 71.35%; when integrated into *AutoAnnotator*, the sentiment accuracy rises to 73.12% (+4.02%) and the toxicity accuracy rises to 77.56% (+6.21%). Besides, compared with direct LLM annotation, *AutoAnnotator* significantly reduces the number of LLM calls (60%+ for sentiment tasks and 70%+ for toxicity tasks). It is worth noting that *AutoAnnotator* not only outperforms existing API models in terms of performance, but also far exceeds them in terms of annotation cost and efficiency (see below). We provide some samples that *AutoAnnotator* can annotate correctly, but API models annotate incorrectly in Figure 4. In addition, we compare *AutoAnnotator* with API models of few-shot settings. Specifically, we use GPT-3.5-turbo to test the performance of zero-shot, three-shot, and five-shot settings on the JPP-Sentiment dataset. The results are shown in Table 4. As expected, increasing the sample size in prompts from zero to five does indeed improve the accuracy of GPT-3.5-turbo from 72.91% to 73.70%. However, while the five-shot baseline (73.70%) is more competitive, our *AutoAnnotator* still significantly outperforms this stronger baseline, which further demonstrates the effectiveness of our method. Further-

more, using a five-shot setting significantly increases API call costs. This contradicts *AutoAnnotator*'s core objective of reducing costs.

**Comparison with LLM Majority-Vote.** We additionally evaluate majority voting across multiple LLMs. As shown in Table 3, *AutoAnnotator*+Minimax needs only $10,643$ API calls to achieve 74.59% sentiment accuracy, outperforming both open-source and API-voting baselines while reducing LLM calls by over 93%. Regardless of whether we ensemble multiple open-source or API LLMs—even with zero-shot or one-shot voting—*AutoAnnotator* consistently outperforms all voting schemes while using far fewer LLM calls.

**Comparison with Mixture-of-Agents (Wang et al., 2024).** In this subsection, we compare *AutoAnnotator* with Mixture-of-Agent (MoA). MoA involves collaboration between multiple powerful LLMs, rather than dedicated SLMs, to solve complex tasks through multiple iterations. In contrast, *AutoAnnotator* facilitates collaboration between two different types of models: powerful LLMs and dedicated SLMs. To verify the effectiveness of our proposed method, we conduct experiments on JPP-Sentiment dataset and compare it with MoA. As shown in Table 5, MoA achieves an accuracy of 72.18%, which is far lower than our method. The reason for MoA's poor performance lies in the fact that it is composed of multiple LLMs. Figure 1(a) clearly shows that on fine-grained semantic understanding tasks such as sentiment classification, the accuracy of general-purpose LLMs (such as Llama3-70B, QwQ-32B, etc.) is significantly lower than that of specially fine-tuned SLMs (such as RoBERTa, BERTweet). Therefore, the MoA framework essentially integrates multiple models that perform poorly on this specific task. While integration can bring some performance improvement, it does not outperform our method.

**Extend *AutoAnnotator* to more tasks.** To verify the versatility of *AutoAnnotator* beyond sentiment and toxicity classification, we conduct additional experiments in the more challenging domain of mathematical reasoning. Specifically, we select two widely used benchmarks, MathQA (Amini et al., 2019) and AQUA-RAT (Ling et al., 2017), for

Table 6: Comparison of the proposed *AutoAnnotator* with state-of-the-art baselines on math tasks.

| Model | Acc on MathQA | Acc on AQUA-RAT |
|---|---|---|
| Mathstral-7B-v0.1 | 39.60% | 36.83% |
| Qwen2.5-Math-7B-Instruct | 80.29% | 79.87% |
| DeepSeek-math-7b-Instruct | 66.43% | 65.33% |
| Self-Consistency (GPT-3.5-turbo) | 65.00% | 64.20% |
| DynaThink+Self-Consistency (GPT-3.5-turbo) | 68.00% | 61.50% |
| *AutoAnnotator* | 85.00% | 83.39% |

the annotation task. Following the setup of our framework, we use three publicly available mathematical models as SLMs for the task-specialist layer: Mathstral-7B-v0.1[4], Qwen2.5-Math-7B-Instruct (Yang et al., 2024b), and DeepSeek-math-7b-Instruct[5]. We compare the performance of *AutoAnnotator* with two strong baselines: (1) the performance of each specialist SLM individually, and (2) the advanced GPT-3.5-turbo plus advanced inference strategies, including self-consistency and DynaThink + self-consistency. As shown in Table 6, *AutoAnnotator* achieves the highest accuracy on both datasets, scoring 85.00% on MathQA and 83.39% on AQUA-RAT. This performance significantly surpasses the strongest single SLM and far exceeds the results achieved using only LLM. These experimental results further demonstrate that the proposed method is a general approach that can improve annotation quality even in complex domains.

**Scenarios where no pre-trained SLM exists.** In this paper, we assume that LLMs can download pre-trained SLMs from Hugging Faces. However, for some niche tasks, these high-quality, existing SLMs may not exist. To address the feasibility of this "cold start", we design the following experiments to simulate this scenario. We use the JPP-

Table 7: Performance comparison of *AutoAnnotator* with and without pre-trained SLMs.

| Model | Scene | Acc |
|---|---|---|
| GPT-3.5-turbo | Downloaded from huggingface | 78.56% |
| | Trained using self-labeled data | 78.29% |

Sentiment dataset as an example to compare two cases. In this cold start scenario, we first manually label 2000 samples from the JPP-Sentiment dataset to form a small seed dataset. Next, we use this seed dataset

---

[4]https://huggingface.co/mistralai/Mathstral-7B-v0.1
[5]https://huggingface.co/deepseek-ai/deepseek-math-7b-instruct

Table 8: Comparison of annotation cost and efficiency between API models and *AutoAnnotator*.

| Model | Token (Input+Output) | GPU Memory | Time Cost | Time Reduction | Cost | Cost Reduction |
|---|---|---|---|---|---|---|
| Deepseek-V3 | 88023 | - | 71.19 minutes | - | 0.027202 $ | - |
| *AutoAnnotator*+Deepseek-V3 | 25629 | 4458 MB | 26.06 minutes | 63.40% | 0.008085 $ | 70.28% |
| Deepseek-R1 | 356532 | - | 212.93 minutes | - | 0.650334 $ | - |
| *AutoAnnotator*+Deepseek-R1 | 109666 | 4458 MB | 39.40 minutes | 81.50% | 0.211581 $ | 67.47% |
| Minimax | 91724 | - | 93.70 minutes | - | 0.012723 $ | - |
| *AutoAnnotator*+Minimax | 19357 | 4458 MB | 30.67 minutes | 67.27% | 0.003112 $ | 75.54% |
| GPT-3.5-turbo | 88514 | - | 34.17 minutes | - | 0.048423 $ | - |
| *AutoAnnotator*+GPT-3.5-turbo | 22235 | 4458 MB | 17.75 minutes | 48.05% | 0.012519 $ | 74.15% |
| GPT-4 | 88027 | - | 30.56 minutes | - | 2.751180 $ | - |
| *AutoAnnotator*+GPT-4 | 23843 | 4458 MB | 18.34 minutes | 40.00% | 0.754470 $ | 72.58% |
| GPT-4o | 87915 | - | 23.20 minutes | - | 0.246540 $ | - |
| *AutoAnnotator*+GPT-4o | 22087 | 4458 MB | 15.21 minutes | 34.44% | 0.064300 $ | 73.92% |

Table 9: Detailed comparison of the time cost of annotation and fine-tuning with SLMs and the time cost of annotation with LLMs.

| SLMs | | | | LLMs | |
|---|---|---|---|---|---|
| Model | Labeling | Finetuning | Total | Model | Labeling |
| cardiffnlp/twitter-roberta-base-sentiment-latest | 0.32 minutes | 0.54 minutes | 0.86 minutes | Deepseek-V3 | 71.19 minutes |
| cardiffnlp/twitter-xlm-roberta-base-sentiment | 0.38 minutes | 0.66 minutes | 1.04 minutes | Deepseek-R1 | 212.93 minutes |
| finiteautomata/bertweet-base-sentiment-analysis | 0.30 minutes | 0.58 minutes | 0.88 minutes | Minimax-abab6.5s-chat | 93.70 minutes |
| s-nlp/roberta_toxicity_classifier | 0.29 minutes | 0.56 minutes | 0.85 minutes | GPT-3.5-turbo | 34.17 minutes |
| JungleLee/bert-toxic-comment-classification | 0.29 minutes | 0.52 minutes | 0.81 minutes | GPT-4o | 23.20 minutes |
| garak-llm/toxic-comment-model | 0.23 minutes | 0.35 minutes | 0.58 minutes | GPT-4 | 30.56 minutes |

to fine-tune 3 SLMs and treat them as experts in the task-specialist layer. Finally, we utilize GPT-3.5-turbo as the meta-controller to run the complete *AutoAnnotator* process. As shown in Table 7, the cold start strategy achieves similar annotation accuracy to the original strategy, further demonstrating the feasibility and robustness of *AutoAnnotator*, even for niche tasks without pre-trained SLMs.

**The efficiency and cost of *AutoAnnotator*.** To evaluate the annotation cost and efficiency of API models and *AutoAnnotator*, we conduct a quantitative analysis from three dimensions: computing resource consumption (the number of tokens and GPU memory usage), annotation time cost, and economic cost. All experiments are performed on NVIDIA A100 GPUs, and the annotation task scale is uniformly set to 1000 samples. As for our *AutoAnnotator*, we set the $\beta = 200$. We conduct experiments on Deepseek-V3, Deepseek-R1, Minimax, GPT-3.5-turbo, GPT-4 and GPT-4o, respectively. As shown in Table 8, *AutoAnnotator* reduces the annotation time by 34.44% (GPT-4o) to 81.50% (Deepseek-R1), with an average reduction of 55.85%. Besides, the annotation cost is reduced by 75.54% (Minimax) at the highest and 67.47% (Deepseek-R1) at the lowest, with an average saving of 72.32%. In Table 8, we provide the annotation time cost for the whole pipeline of *AutoAnnotator*. Herein, we further provide the annotation time cost for SLMs as well as their fine-tuning cost. For a fair comparison, we annotate and fine-tune on the same 1000 difficult samples. As shown in Table 9, compared to LLMs, SLMs require significantly less time for annotation and fine-tuning, typically under 1 minute in total per model. In contrast, LLM-based annotation is substantially more time-consuming, with labeling times ranging from 23 to over 200 minutes depending on the model. This highlights the practical advantage of using SLMs for efficient labeling and fine-tuning in real-world scenarios. In general, *AutoAnnotator* has achieved a significant improvement in annotation efficiency and a significant reduction in annotation costs while maintaining annotation quality through the LLMs and SLMs collaborative annotation paradigm.

## 4.3 Ablation Study

**The number of SLMs used in the task-specialist layer.** To explore the impact of the number of SLMs on the annotation performance, we perform ablations on the JPP-Sentiment dataset using GPT-3.5-turbo as the meta-controller LLM. We vary the number of SLMs $k$ participating in the multi-model consensus annotation from 2 to 5 and report the final annotation accuracy in Table 10. We find that the annotation

Table 10: The impact of the number of SLMs used for annotation on the annotation performance.

| k | 2 | 3 | 4 | 5 |
|---|---|---|---|---|
| Acc | 73.32% | 78.56% | 76.26% | 76.26% |

Table 11: The impact of the number of hard samples used for continuous fine-tuning at each stage on the final performance.

| $\beta$ | 500 | 1000 | 2000 | 3000 |
|---|---|---|---|---|
| Acc | 76.17% | 73.91% | 78.56% | 73.12% |

performance is best when $k = 3$. We believe that the reason for the decline in annotation performance when $k$ is greater than 3 is that when there are too many SLMs, the performance differences between them will cause the poorly performing SLMs to affect the voting results, ultimately affecting the annotation effect. Nevertheless, *AutoAnnotator* is flexible enough that it can still operate if the meta-controller layer finds only one or two relevant SLMs after searching. Therefore, considering the computational cost and annotation accuracy, we use $k = 3$ as the default in this paper.

**The number of samples in the hard sample pool $\mathcal{D}_{\mathbf{hard}}$.** To explore the impact of the number of hard samples used for continuous fine-tuning at each stage on the annotation performance, we perform ablations on the JPP-Sentiment dataset using GPT-3.5-turbo. We set the sample size $\beta$ to $\{500, 1000, 2000, 3000\}$. As shown in Table 11, we find that the annotation performance peaks at $\beta = 2000$, therefore, we adopt $\beta = 2000$ as the default hard-sample batch size in this paper.

**The annotation accuracy between SLMs and LLMs on the hard sample pool.** To verify that LLMs indeed improve annotation quality, we report the accuracy on JPP-Sentiment dataset of SLMs and LLMs for samples in the hard sample pool, using gold labels for evaluation. As shown in Table 12, LLMs consistently outperform SLMs on difficult samples, achieving notably higher annotation accuracy. This also explains why our framework can provide high-quality responses for continual training without human intervention.

Table 12: The annotation accuracy between SLMs and LLMs on the hard sample pool.

| Number of Difficult Samples | Model | Acc |
|---|---|---|
| 2881 | cardiffnlp/twitter-roberta-base-sentiment-latest | 58.83% |
| | cardiffnlp/twitter-xlm-roberta-base-sentiment | 42.97% |
| | finiteautomata/bertweet-base-sentiment-analysis | 57.20% |
| | GPT-3.5-turbo | 65.29% |
| 2836 | cardiffnlp/twitter-roberta-base-sentiment-latest | 58.39% |
| | cardiffnlp/twitter-xlm-roberta-base-sentiment | 42.77% |
| | finiteautomata/bertweet-base-sentiment-analysis | 56.52% |
| | Minimax-abab6.5s-chat | 68.48% |
| 3148 | cardiffnlp/twitter-roberta-base-sentiment-latest | 61.75% |
| | cardiffnlp/twitter-xlm-roberta-base-sentiment | 47.11% |
| | finiteautomata/bertweet-base-sentiment-analysis | 61.50% |
| | Deepseek-V3 | 69.85% |

**Voting using only SLMs.** To verify that the performance gains of *AutoAnnotator* do not come solely from simple SLM ensembles, we introduced a "SLMs Voting" baseline. Specifically, we use the same 3 SLMs, employ a simple majority voting mechanism for data labeling on the JPP-Sentiment dataset, and do not include secondary review by LLMs or expert-guided iterative optimization. As shown in Table 13, SLMs voting even performs worse than the best single SLM. This shows that simple integrated voting can be polluted by poor-performing SLMs (such as SLM2), leading to a decline in overall performance. In comparison, *AutoAnnotator* achieves better results, demonstrating the importance of secondary review and expert-guided iterative optimization.

**How to select SLMs.** In this paper, we use an LLM as the meta-controller to adaptively select SLMs. Here, we conduct additional experiments using existing model selection methods. Specifically, we choose a classic and effective model selection algorithm, LogME (You et al., 2021), and use it for model recommendation on the JPP-Sentiment dataset. Since LogME requires some models to be given in advance, we select 10 models for sentiment classification from Huggingface, including the three SLMs used in our paper. We run the LogME algorithm to evaluate and recommend the three best SLMs from a pool of 10 models. Then, we replaced the models recommended by LLM with the SLMs recommended by LogME for data labeling.

Table 13: Comparison of the proposed *AutoAnnotator* with SLMs Voting.

| Method | Acc | Method | Acc |
|--------|-----|--------|-----|
| SLM1 | 70.31% | *AutoAnnotator*+Minimax | 81.20% |
| SLM2 | 59.76% | *AutoAnnotator*+Deepseek-V3 | 77.96% |
| SLM3 | 69.22% | *AutoAnnotator*+GPT-3.5-turbo | 78.56% |
| SLMs Voting | 67.76% | *AutoAnnotator*+Human | 82.83% |

Table 14: Ablation experiments on different SLM selection methods.

| Method | Acc |
|--------|-----|
| Recommended by LogME | 74.57% |
| Recommended by LLM | 78.56% |

As shown in Table 14, compared to SLMs recommended by LogME (74.57%), SLMs recommended by LLM achieves a higher annotation accuracy (78.56%) under the *AutoAnnotator* framework. This demonstrates that LLM's powerful reasoning and generalization capabilities enable it to form a more comprehensive and profound understanding of annotation tasks, thus surpassing traditional, single-metric-based strategies in model selection.

## 5 Conclusion

In this paper, we propose a new paradigm for multi-model collaborative annotation and designs a fully automatic annotation framework *AutoAnnotator* based on it. Specifically, *AutoAnnotator* consists of a meta-controller layer and a task-specialist layer. Specifically, the meta-controller layer is responsible for recommending appropriate annotation SLMs, generating the code required for annotation, and rechecking difficult samples that cannot be determined by SLMs. while the task-specialist layer is responsible for the actual annotation. To enhance the generalization of the SLMs, we use the difficult samples obtained from the second verification of the LLM as a reinforcement learning set, and periodically send it to the SLMs for continuous fine-tuning. Extensive experiments demonstrate the effectiveness of *AutoAnnotator* on six datasets.

**Limitations**

While promising, there are still some drawbacks of *AutoAnnotator*. The model selection, code generation, and difficult sample review of the entire framework are all driven by LLMs. Therefore, the performance of *AutoAnnotator* depends to a certain extent on the quality of LLMs.

**Acknowledgments**

This work was partially supported by the Key R&D Program of Zhejiang under Grant 2022C01018 and 2024C01025, by the National Key R&D Program under Grant 2025YFA1510900 and 2025YFA1510902, by the National Natural Science Foundation of China under Grant U21B2001, 62301492 and 61973273, by the Baima Lake Laboratory Joint Fund of Zhejiang Provincial Natural Science Foundation of China under Grant LBMHZ25F020002 and by Key Technology Research and Development Program Project of Hangzhou under Grant 2024SZD1A23 and 2025SZD1A41.

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
