# Appendix

## Organization of the Appendix

The Appendix is organized as follows.

- Section A introduces the dataset we used.

- Section B provides the prompts we used.

- Section C provides the codes generated by LLMs.

## A   Dataset Description

Here, we introduce the two types of datasets used in this study.  As for sentiment classification tasks, three public datasets are used, namely **mteb/tweet_sentiment_extraction**[6], **jppgks/twitter-financial-news-sentiment**[7] and **LYTinn/sentiment-analysis-tweet**[8].  As for toxicity classification tasks, three public datasets are used, namely **karthikarunr/Cyberbullying-Toxicity-Tweets**[9], **jiaxin-wen/Implicit-Toxicity**[10] and **heegyu/toxic_conversations_balanced**[11].

- **mteb/tweet_sentiment_extraction**: $24,739$ samples from its training set (train-00000-of-00001.parquet) are split into a 7:3 ratio, yielding $17,317$ training and $7,422$ test samples.

- **jppgks/twitter-financial-news-sentiment**: The full training set ($9,543$ samples) and test set ($2,388$ samples) are directly adopted as the framework's training and test sets.

- **LYTinn/sentiment-analysis-tweet**: The entire training set ($11,536$ samples) and test set ($3,377$ samples) are directly adopted as the framework's training and test sets.

- **karthikarunr/Cyberbullying-Toxicity-Tweets**: $23,005$ samples from its training set (train-00000-of-00001.parquet) are split into $16,104$ training and $6,901$ test samples (7:3 ratio).

- **jiaxin-wen/Implicit-Toxicity**: $22,426$ samples from train/sft-train.json are divided into $15,698$ training and $6,728$ test samples (7:3 ratio).

- **heegyu/toxic_conversations_balanced**: $23,820$ samples from train.csv are split into $16,674$ training and $7,146$ test samples (7:3 ratio).

## B   Prompts

We provide the prompts used in the paper in Table A and Table B.

## C   Codes Generated by LLMs

We provide the codes generate by LLMs below.

---

[6]https://huggingface.co/datasets/mteb/tweet_sentiment_extraction
[7]https://huggingface.co/datasets/jppgks/twitter-financial-news-sentiment
[8]https://huggingface.co/datasets/LYTinn/sentiment-analysis-tweet
[9]https://huggingface.co/datasets/karthikarunr/Cyberbullying-Toxicity-Tweets
[10]https://huggingface.co/datasets/jiaxin-wen/Implicit-Toxicity
[11]https://huggingface.co/datasets/heegyu/toxic_conversations_balanced

| System Prompt | |
| --- | --- |
| Model Selection | Now I need you to help me write a code. Note that you need to strictly follow my requirements and the content you generate should be directly runnable code.
Requirements:
1. Model search: Find text annotation models similar to BERT on Hugging Face according to the text annotation requirements of sentiment classification (only supporting positive, negative, neutral and their respective confidence scores) and toxic content detection (only supporting toxic and its confidence score). Note that models with multiple labels like unitary/toxic-bert do not meet the requirements.
2. Quantity requirements: 3 sentiment classification models and 3 toxic content detection models.
3. After selection, set up a UI interface for me to view the information of the selected models (using tkinter).
4. Save the table in the UI interface to the local path {config.path}.
Note:
1. The table is for reference only. The number of models, model parameters, and HF downloads are inaccurate and need to be investigated by you.
2. You need to search for models that meet the conditions and remember them. Your code only includes the UI part (including models that meet the conditions).
3. Please strictly follow the above requirements when writing the code.
4. Your answer should only contain the code and nothing else. |
| Model Deployment | Now I need you to help me write a code. Note that you need to strictly follow my requirements and the content you generate should be directly runnable code. I need to deploy a Hugging Face model locally. The model address is https://hf-mirror.com/models. Please help me complete the following tasks:
1. Model ID acquisition: Get the model IDs I want to download from the 'Model ID' column in the file {Local File}.
2. Save path: I want to save the models to the local directory {Local Path}. For example, save 'cardiffnlp_twitter-roberta-base-sentiment-latest' to {Local Path}.
3. Code generation: Please generate a complete and directly runnable Python code to download the models locally. Please generate the code according to the above requirements to complete the deployment.
Note: Your answer should only contain the code and nothing else. |
| Data Annotation | Now I need you to help me write a code. Note that you need to strictly follow my requirements and the content you generate should be directly runnable code. Your task is to help me call a pre-trained BERT model (or similar models) from the local and use this model to label text data. The following are the specific requirements of the task:
Task description:
1. Model type: Use a pre-trained model like BERT or similar ones (e.g., RoBERTa, DistilBERT, etc.).
2. Model ID: {Model ID}.
3. Local model address: {Local Path}.
4. Task type: {Task Type}.
5. Input data: The address of the data file in the first column of the local xlsx file is {Local File}.
6. Output result: The labeled result. If there is a confidence score, it is also required. Save it to {Local File}.
7. Requirements for the result file: Change the column name of the label column in the saved file to {label_col_name}, and change the column name of the confidence score column to {confidence_col_name}'. |
| Model Fine-tuning | Now I need you to help me write a code. Note that you must strictly follow my requirements, and the content you generate should be directly runnable code. Your task is to write code to perform full-parameter fine-tuning on the {Model ID} model. The following are the specific requirements of the task:
Task description:
1. Model type: Use a pre-trained model like BERT or similar ones (e.g., RoBERTa, DistilBERT, etc.).
2. Model ID: {Model ID}.
3. Local model address: {Local Path}.
4. Fine-tuning data: The 'text' column in the {Local File} is the text column of the fine-tuning data, and the 'label' column is the label column of the fine-tuning data.
5. Save address for the fine-tuned model: {Local Path}.
Note: Your answer should only contain the code and nothing else. |

Table A: System Prompt for Data Annotation.

| Prompt | |
| --- | --- |
| Sentiment Classification | You are an autoclassifier that's responsible for labeling input text. You must respond with only one of these labels: positive, negative, neutral. |
| Toxicity Classification | You are an autoclassifier that's responsible for labeling input text. You must respond with only one of these labels: toxic, non-toxic. |

Table B: Prompt for Data Annotation.

**Algorithm 1:** PyTorch implementation of model selection generated by LLMs.

```python
import tkinter as tk
from tkinter import ttk
import pandas as pd
import os

# Selected models based on the requirements
models = [
    {"Serial Number": 1, "Model ID": "cardiffnlp/twitter-roberta-base-sentiment-latest",
     "Type": "sentiment", "Parameters": "125M", "HF Downloads": "2.43M"},
    {"Serial Number": 2, "Model ID": "cardiffnlp/twitter-xlm-roberta-base-sentiment",
     "Type": "sentiment", "Parameters": "125M", "HF Downloads": "2.06M"},
    {"Serial Number": 3, "Model ID": "finiteautomata/bertweet-base-sentiment-analysis",
     "Type": "sentiment", "Parameters": "110M", "HF Downloads": "1.06M"},
    {"Serial Number": 4, "Model ID": "s-nlp/roberta_toxicity_classifier",
     "Type": "toxic", "Parameters": "110M", "HF Downloads": "160K"},
    {"Serial Number": 5, "Model ID": "JungleLee/bert-toxic-comment-classification",
     "Type": "toxic", "Parameters": "110M", "HF Downloads": "46.3K"},
    {"Serial Number": 6, "Model ID": "garak-llm/toxic-comment-model",
     "Type": "toxic", "Parameters": "110M", "HF Downloads": "9.19K"}
]

# Convert to DataFrame
df = pd.DataFrame(models)

# Function to save the table to Excel
def save_to_excel():
    save_path = 'C:\\Users\\21888\\Desktop\\1-24\\model.xlsx'
    df.to_excel(save_path, index=False)
    status_label.config(text=f"Table saved to {save_path}")

# Create the main window
root = tk.Tk()
root.title("Hugging Face Model Search")

# Create a frame for the table
frame = ttk.Frame(root)
frame.pack(pady=20)

# Create a treeview to display the table
tree = ttk.Treeview(frame, columns=list(df.columns), show="headings")
for col in df.columns:
    tree.heading(col, text=col)
    tree.column(col, width=100)
tree.pack()

# Insert data into the treeview
for index, row in df.iterrows():
    tree.insert("", "end", values=list(row))

# Create a button to save the table
save_button = ttk.Button(root, text="Save to Excel", command=save_to_excel)
save_button.pack(pady=10)

# Status label
status_label = ttk.Label(root, text="")
status_label.pack()

# Run the application
root.mainloop()
```

---

**Algorithm 2:** PyTorch implementation of model deployment generated by LLMs.

---

```python
import pandas as pd
from transformers import AutoTokenizer, AutoModel
import os

def get_model_ids():
    file_path = r'C:\Users\21888\Desktop\1-24\model.xlsx'
    df = pd.read_excel(file_path)
    model_ids = df['Model ID'].tolist()
    return model_ids

def download_and_save_models(model_ids, save_base_dir):
    for model_id in model_ids:
        try:
            tokenizer = AutoTokenizer.from_pretrained(model_id)
            model = AutoModel.from_pretrained(model_id)

            model_save_dir = os.path.join(save_base_dir, model_id.replace("/", "_"))
            os.makedirs(model_save_dir, exist_ok=True)

            tokenizer.save_pretrained(model_save_dir)
            model.save_pretrained(model_save_dir)
            print(f"model {model_id} has been saved to {model_save_dir}")
        except Exception as e:
            print("error")

if __name__ == "__main__":
    model_ids = get_model_ids()

    save_base_dir = r'E:\model'

    download_and_save_models(model_ids, save_base_dir)
```

---

**Algorithm 3:** PyTorch implementation of data annotation generated by LLMs.

---

```python
import torch
import pandas as pd
from transformers import AutoTokenizer, AutoModelForSequenceClassification

model_path = r'E:\model\cardiffnlp_twitter-roberta-base-sentiment-latest'
tokenizer = AutoTokenizer.from_pretrained(model_path)
model = AutoModelForSequenceClassification.from_pretrained(model_path)

input_file = r'C:\Users\21888\Desktop\test\11.xlsx'
df = pd.read_excel(input_file, header=None)

df = df.dropna(subset=[0])

texts = df[0].astype(str).tolist()

results = []

for text in texts:
    inputs = tokenizer(text, return_tensors='pt', truncation=True, padding=True)
    with torch.no_grad():
        logits = model(**inputs).logits
    probabilities = torch.softmax(logits, dim=-1)
    sentiment_label = torch.argmax(probabilities, dim=-1).item()
    sentiment_confidence = probabilities[0][sentiment_label].item()
    results.append({'sentiment_1': sentiment_label, 'sentiment_confidence_1': sentiment_confidence})

output_df = pd.DataFrame(results)
output_file = r'C:\Users\21888\Desktop\test\result_sentiment_1.xlsx'
output_df.to_excel(output_file, index=False)
```

---

---

**Algorithm 4:** PyTorch implementation of model fine-tuning generated by LLMs.

---

```python
import torch
from transformers import RobertaForSequenceClassification, RobertaTokenizer, Trainer, TrainingArguments
from sklearn.model_selection import train_test_split
import pandas as pd
from datasets import Dataset

data = pd.read_excel('C:\\Users\\21888\\Desktop\\test\\finetune_sentiment.xlsx')
texts = data['text'].tolist()
labels = data['label'].tolist()

train_texts, val_texts, train_labels, val_labels = train_test_split(texts, labels, test_size=0.2, random_state=42)

train_dataset = Dataset.from_dict({'text': train_texts, 'label': train_labels})
val_dataset = Dataset.from_dict({'text': val_texts, 'label': val_labels})

model_name = 'E:\\model\\cardiffnlp_twitter-roberta-base-sentiment-latest'
model = RobertaForSequenceClassification.from_pretrained(model_name)
tokenizer = RobertaTokenizer.from_pretrained(model_name)

def preprocess_function(examples):
    return tokenizer(examples['text'], truncation=True, padding='max_length', max_length=128)

train_dataset = train_dataset.map(preprocess_function, batched=True)
val_dataset = val_dataset.map(preprocess_function, batched=True)

training_args = TrainingArguments(
    output_dir='E:\\fine-tune-model\\cardiffnlp_twitter-roberta-base-sentiment-latest',
    evaluation_strategy="epoch",
    learning_rate=2e-5,
    per_device_train_batch_size=16,
    per_device_eval_batch_size=16,
    num_train_epochs=3,
    weight_decay=0.01,
    save_total_limit=2,
    save_steps=500,
    logging_dir='./logs',
    logging_steps=10,
)

trainer = Trainer(
    model=model,
    args=training_args,
    train_dataset=train_dataset,
    eval_dataset=val_dataset,
)

trainer.train()

trainer.save_model('E:\\fine-tune-model\\cardiffnlp_twitter-roberta-base-sentiment-latest')

tokenizer.save_pretrained('E:\\fine-tune-model\\cardiffnlp_twitter-roberta-base-sentiment-latest')
```

---