# OpenReview forum: "AutoAnnotator: A Collaborative Annotation Framework for Large and Small Language Models"
_TMLR — Accepted by TMLR_

### Review · Reviewer_yg5N · 2025-09-14

**Summary Of Contributions:**

The paper proposes an auto labeling framework tailored to two downstream tasks, sentiment analysis and toxicity classification, which reduces inference costs by iteratively fine-tuning openly available SLMs for LLM-labelled data points with prediction disagreements.

The approach is evaluated for the two tasks on selected datasets and compared to individual SLM performance, standard majority voting, open source LLMs and APIs.

The results show that the LLM-guided approach is partially better compared to the best individual SLMs. It is better than general LLMs and APIs though.

**Audience:**

Yes

**Audience Explanation:**

- In general, the paper's goals and possible findings are of interest for sure, as they would support reducing API costs and improving predictive performance. I believe, however, that the evaluation and presentation of the paper needs to be improved for that.

**Broader Impact Concerns:**

-

**Claims And Evidence:**

No

**Claims Explanation:**

- The task layer problem is strongly related and could partially be reduced to classical expert advice / mixture of experts / boosting, but no baseline is used at all (except for standard majority voting). This literature is not discussed in the related works. It is also missing as baselines for the empirical evaluation.
- Also, the approach should be compared to classical teacher student architecture / knowledge distillation.
- There are also more general solutions for LLM available such as Mixture-of-Agents (https://openreview.net/pdf?id=h0ZfDIrj7T)
- The paper title suggest a general annotation framework, but the evaluation only targets sentiment analysis and toxicity classification. This seems like an overstatement
- Even though ablation study was done, the approach seems very dependent on hyperparameter k being small, which feels like the majority voting might not be enough
- The Meta-Controller Layer is not well-evaluated. It seems the SLMs were chosen by the layer, but it is unclear how many other options would be available (i.e., how good this choice is globally) or how difficult this task even is. Here, it would be important to check if selection diversity wrt predicted classes is important.
- The evaluation results for the LLM-guided SLM fine-tuning are not always better than using individual SLMs. This increases importance to test simple weighting of SLMs based on their performances on held out validation data. There are prominent model selection methods one could also compare to (based on a pool of "general" options for the task)

**Requested Changes:**

Please change the manuscript and evaluation based on the mentioned shortcomings above plus by answering these questions:
- How does the approach related to mentioned foundational works, e.g., expert advice / mixture of experts / teacher-student architectures?
- How does the approach related to mixture of agents?
- Do you have an idea why k=3 is performing the best? Why can the system not efficiently use more than 3 SLMs? Maybe the 5 chosen SLMs are not diverse enough, so individual "brilliant" SLMs are not strongly weighted enough?
- Are the combinations of SLMs of size 3 which might not work together in your approach?
- Why is not possible to compare against some of the mentioned LLM combination papers you mention in your related works?
- What are the theoretical requirements for the SLM performances? If no disagreement occurs and the SLMs perform bad, you never improve.
- What if one gives an LLM or API some more few-shot examples based on a held-out validation set? Could this be more competitive?
- In the one-shot setting, is there really only one training example passed in-context?

---

> ### Author Response · Authors · 2025-11-05
> **Responding to the Reviewer yg5N‘s comments (1)**
>
> Thank you very much for your valuable insights, which are essential for us to improve and refine this work. We have carefully considered each of your questions and will provide detailed responses and explanations to each of your comments below.
>
> **How does the approach related to mentioned foundational works, e.g., expert advice / mixture of experts / teacher-student architectures? This literature is not discussed in the related works. It is also missing as baselines for the empirical evaluation.** Thank you very much for your valuable feedback. We would like to clarify the key differences between our AutoAnnotator framework and these classic concepts:
> - Difference from expert advice. Although our paper involves secondary review using LLM, this is only a small part of our paper.
> - Difference from mixture of experts. Although both approaches use multiple models for experimentation, their core concepts, architecture, and goals are fundamentally different. We elaborate on these differences in a later response.
> - Difference from "teacher-student architectures". The ultimate goal of the teacher-student architecture is to train a smaller, more powerful model (student) that mimics the behavior of the larger model (teacher). However, the ultimate goal of AutoAnnotator is to produce high-quality labeled datasets. While we may improve the performance of SLMs through iterative optimization, this is merely an intermediate step towards achieving more efficient and accurate annotation, not the final objective.
>
> Besides, we have included content related to multi-agent systems in the related work section and more baselines in Section 4.
>
> **The approach should be compared to classical teacher student architecture / knowledge distillation.** Thank you for comparing our work to teacher student architecture /  knowledge distillation. While both involve the interaction of a "large model" and a "small model," our AutoAnnotator differs fundamentally from classic KD:
>   - In knowledge extraction, LLM is the "teacher," possessing more knowledge, and its goal is to pass on its soft labels to the "students," enabling them to emulate the teacher's behavior and acquire stronger abilities. Furthermore, the ultimate goal of knowledge distillation is to improve the performance of small models.
> - In AutoAnnotator, SLMs are considered domain experts and are the main force in data annotation. LLMs act as "meta-controllers" and are responsible for (1) selecting SLMs for annotation and (2) conducting secondary reviews when there are disagreements in the SLM voting. The ultimate goal of AutoAnnotator is to produce high-quality labeled datasets. While we also improve the performance of SLM through "iterative optimization," this is only an intermediate process aimed at making SLM perform the labeling task more accurately.
>
> **There are also more general solutions for LLM available such as Mixture-of-Agents.** Thank you for your insightful suggestions. MoA frameworks often involve collaboration among multiple LLMs, resolving complex tasks through multiple iterations. To verify the effectiveness of our proposed method, we compare it with MoA. As shown in Table 1, on the JPP-Sentiment dataset, a MoA baseline composed of multiple LLMs (Llama3.1-8B-Instruct, Qwen2.5-7B-Instruct, Mistral-7b-v0.2) achieves an accuracy of 72.18%. The reason for MoA's poor performance lies in the fact that it is composed of multiple LLMs. Figure 1(a) clearly shows that on fine-grained semantic understanding tasks such as sentiment classification, the accuracy of general-purpose LLMs (such as Llama3-70B, QwQ-32B, etc.) is significantly lower than that of specially fine-tuned SLMs (such as RoBERTa, BERTweet). Therefore, the MoA framework essentially integrates multiple models that perform poorly on this specific task. While integration can bring some performance improvement, it does not outperform our method. We have also included comparative experiments and related analyses with MoA in the revised version of the paper.
> | Model | Acc |
> | :--- | :--- |
> | MoA | 72.18% |
> | Ours + Human | 82.83% |
> | Ours + Minimax | 81.20% |
> | Ours + Deepseek-V3 | 77.96% |
> | Ours + GPT-3.5-turbo | 78.56% |

---

> > ### Author Response · Authors · 2025-11-05
> > **Responding to the Reviewer yg5N‘s comments (2)**
> >
> > **The paper title suggest a general annotation framework, but the evaluation only targets sentiment analysis and toxicity classification.** Thank you very much for your valuable feedback. Based on your suggestion, we have added experiments on more challenging mathematical reasoning tasks to verify whether our framework is applicable to more complex domains. Specifically, we select two widely used mathematical reasoning datasets, MathQA and AQUA-RAT, as the annotation task, and three publicly available models, Mathstral-7B-v0.1, Qwen2.5-Math-7B-Instruct, and DeepSeek-math-7b-Instruct, as the SLMs. To validate the effectiveness of our method, we compare the performance of AutoAnnotator with that of individual SLMs, as well as with GPT-3.5-turbo with additional inference strategies such as Self-Consistency and DynaThink. As shown in the table below, AutoAnnotator significantly outperforms all baselines, further demonstrating its generalization ability.
> >
> > | Model | Acc on MathQA | Acc on AQUA-RAT |
> > | :--- | :--- | :--- |
> > | Mathstral-7B-v0.1 | 39.60% | 36.83% |
> > | Qwen2.5-Math-7B-Instruct | 80.29% | 79.87% |
> > | DeepSeek-math-7b-Instruct | 66.43% | 65.33% |
> > | Self-Consistency (GPT-3.5-turbo) | 65.00% | 64.20% |
> > | DynaThink+Self-Consistency (GPT-3.5-turbo) | 68.00% | 61.50% |
> > | **AutoAnnotator** | **85.00%** | **83.39%** |
> >
> > **It seems the SLMs were chosen by the layer, but it is unclear how many other options would be available (i.e., how good this choice is globally) or how difficult this task even is. Here, it would be important to check if selection diversity wrt predicted classes is important.** To address your concern, we have added a comparative experiment that directly compares our model selection using LLM with a well-known, traditional model selection method LogME [2]. The experimental results strongly demonstrate the effectiveness of using LLM as the meta-controller to select SLM. Please see the reply below for detailed experimental information.
> >
> >
> > **Test simple weighting of SLMs based on their performances on held out validation data. There are prominent model selection methods one could also compare to.**  Based on your suggestion, we have added these two experiments. First, we conduct an experiment on the JPP-Sentiment dataset using direct voting from 3 SLMs. As shown in the table below, simple SLM voting is ineffective, poorly performing SLMs (such as SLM2) will pollute the voting results, resulting in a final accuracy that is even worse than the single best model.
> >
> >
> > | Method                          | Acc    |
> > | :------------------------------ | -----: |
> > | SLM1                             | 70.31% |
> > | SLM2                             | 59.76% |
> > | SLM3                             | 69.22% |
> > | SLMs Voting                      | 67.76% |
> > | AutoAnnotator + Minimax          | 81.20% |
> > | AutoAnnotator + Deepseek-V3      | 77.96% |
> > | AutoAnnotator + GPT-3.5-turbo    | 78.56% |
> > | AutoAnnotator + Human            | 82.83% |
> >
> > In this paper, we use an LLM as the meta-controller to adaptively select SLMs. To answer your questions, we conduct additional experiments using existing model selection methods. Specifically, we choose a classic and effective model selection algorithm, LogME, and use it for model recommendation on the JPP-Sentiment dataset. Since LogME requires some models to be given in advance, we select 10 models for sentiment classification from Huggingface, including the three SLMs used in our paper. We run the LogME algorithm to evaluate and recommend the three best SLMs from a pool of 10 models. Then, we replaced the models recommended by LLM with SLMs recommended by LogME for data labeling. As shown in table below, compared to SLMs recommended by LogME (74.57%), SLMs recommended by LLM achieves a higher annotation accuracy (78.56%) under the AutoAnnotator framework.This demonstrates that LLM's powerful reasoning and generalization capabilities enable it to form a more comprehensive and profound understanding of annotation tasks, thus surpassing traditional, single-metric-based strategies in model selection.
> >
> > | Method                      | Acc    |
> > | :-------------------------- | -----: |
> > | Recommended by LogME        | 74.57% |
> > | Recommended by LLM         | 78.56% |

---

> > > ### Author Response · Authors · 2025-11-05
> > > **Responding to the Reviewer yg5N‘s comments (3)**
> > >
> > > **How does the approach related to mixture of agents?** Although both approaches use multiple models for experimentation, their core concepts, architecture, and goals are fundamentally different.
> > > - In terms of goals. MoA leverages multiple LLMs to iteratively enhance the generation quality. AutoAnnotator enables cost-effective and high-quality data annotation.
> > > - In terms of core concepts. MoA involves collaboration between multiple powerful LLMs but does not involve specialized SLMs. AutoAnnotator facilitates collaboration between two different types of models: powerful LLMs and specialized SLMs.
> > > - In terms of architecture. MoA consists of 4 layers. LLMs in the first layer independently generate responses to a given prompt. These responses are then presented to agents in the next layer for further refinement. This iterative refinement process continues for several cycles until obtaining a more robust and comprehensive response. AutoAnnotator consists of two layers. The upper-level meta-controller layer uses the generation and reasoning capabilities of LLMs to select SLMs for annotation, automatically generate annotation code and verify difficult samples; the lower-level task-specialist layer consists of multiple SLMs that perform annotation through multi-model voting.
> > > - In terms of training. MoA is a static framework. It requires no fine-tuning. The entire collaborative process is achieved through prompting engineering. The AutoAnnotator is a dynamic learning framework. SLMs are continuously fine-tuned and improved throughout the process.
> > >
> > > **Do you have an idea why k=3 is performing the best? Why can the system not efficiently use more than 3 SLMs? Maybe the 5 chosen SLMs are not diverse enough, so individual "brilliant" SLMs are not strongly weighted enough?** We believe that the decrease in annotation performance as the k value increases is due to the following reasons: When there are too many SLMs, the performance differences between them can lead to poorly performing SLMs influencing the voting results, ultimately affecting the annotation effect. We believe this is not a problem of insufficient diversity, but rather an inherent characteristic of the "majority voting" mechanism we employ. Because these models have different model architectures and were trained on different training datasets, we believe they possess sufficient "diversity" in terms of knowledge and error patterns.
> > > 	Furthermore, another reason we want to emphasize is that if the number of SLMs is too large, their inference costs and corresponding training costs will also increase, which does not align with the original intention of our low-cost annotation framework.
> > >
> > > **Are the combinations of SLMs of size 3 which might not work together in your approach?** During our experiments, the three SLMs automatically selected from the Hugging Face by the Meta-Controller based on the task characteristics do not fail to work together.
> > >
> > > **Why is not possible to compare against some of the mentioned LLM combination papers you mention in your related works?**  In the related work section, we mention several papers on the collaboration between LLMs and SLMs, such as Xu et al. (2023), CoGenesis, CITER, Collab-RAG and Glocker et al. (2025). We do not use them as benchmarks for comparison because the problems they solve are fundamentally different from those of our AutoAnnotator. Specifically, Xu et al. (2023) aim to improve LLM in-context learning. CoGenesis aims to address privacy concerns logically.  CITER focuses on improving efficiency while ensuring generation quality. Collab-RAG aims to Boost the retrieval-augmented generation for complex question answering. Nevertheless, we fully agree that it is necessary to compare our method with other methods. To this end, based on the reviewers' suggestions, we add a more relevant baseline: Mixture-of-Agents (MoA).

---

> > > > ### Author Response · Authors · 2025-11-05
> > > > **Responding to the Reviewer yg5N‘s comments (4)**
> > > >
> > > > **What are the theoretical requirements for the SLM performances? If no disagreement occurs and the SLMs perform bad, you never improve.** We appreciate the reviewers' insightful comments. We believe it's unlikely that SLMs would be without divergence in real-world applications. As shown in Table 2 of the paper, SLMs automatically selected by Meta-Controller for the task are not single model variants. For example, in the sentiment classification task, it selects RoBERTa, XLM-RoBERTa, and BERTweet; in the toxicity classification task, it selects RoBERTa, BERT, and DistilBERT. These SLMs have different model architectures and are trained by different institutions on different training datasets, resulting in different knowledge blind spots and error patterns. Therefore, the probability that they would "consistently" make the exact same mistakes when encountering a sample is very low. Furthermore, regarding the claim that "SLM performs poorly," we would like to clarify the core motivation and applicable premise of our framework. Our design of the AutoAnnotator is precisely based on the fact that SLM performs exceptionally well in specific domains. As shown in Figure 1(a) of our paper, in tasks requiring fine-grained semantic understanding, such as sentiment classification, specially fine-tuned SLMs (e.g., RoBERTa, BERTweet) achieve significantly higher accuracy than general-purpose LLMs (e.g., GPT-3.5-turbo, Llama3-70B). Therefore, the premise that "SLM performs poorly" does not hold true in our applicable domain.
> > > >
> > > > **What if one gives an LLM or API some more few-shot examples based on a held-out validation set? Could this be more competitive?** Thank you for your insightful suggestion. Adding more examples to the prompts (i.e., few-shot prompting) is indeed a common and effective way to improve LLM performance. We have included both zero-shot and one-shot settings in our paper so far. ​​We conduct additional experiments as you suggested. Specifically, we use GPT-3.5-turbo to test the performance of zero-shot, three-shot, and five-shot settings on the JPP-Sentiment dataset. The results are shown in the table below. As expected, increasing the sample size from one to five does indeed improve the accuracy of GPT-3.5-turbo from 72.91% to 73.70%. While the five-shot baseline (73.70%) is more competitive, our AutoAnnotator framework still significantly outperforms this stronger baseline. Furthermore, the five-shot examples drastically increase the cost of API calls.This contradicts our AutoAnnotator's core goal of significantly reducing costs. Thank you for your valuable feedback. We have already included this few-shot setup experiment in the revised version of the paper.
> > > >
> > > > | Model | Acc |
> > > > | :--- | :--- |
> > > > | GPT-3.5-turbo (zero-shot) | 72.91% |
> > > > | GPT-3.5-turbo (three-shot) | 73.62% |
> > > > | GPT-3.5-turbo (five-shot) | 73.70% |
> > > > | AutoAnnotator+Minimax | 81.20%|
> > > > | AutoAnnotator+Deepseek-V3 |77.96% |
> > > > |AutoAnnotator+GPT-3.5-turbo| 78.56%|
> > > > | AutoAnnotator+Human| 82.83%|
> > > >
> > > >
> > > >
> > > > **In the one-shot setting, is there really only one training example passed in-context?** The one-shot setting we're referring to means that a labeled example is included in the LLMl's input prompt. This setting is the same as that of most papers [1].
> > > >
> > > >
> > > > [1] Large language models are zero-shot reasoners. Advances in neural information processing systems, 35, 22199-22213.
> > > >
> > > > [2] Logme: Practical assessment of pre-trained models for transfer learning. International Conference on Machine Learning, 2021

---

### Review · Reviewer_TGDB · 2025-09-26

**Summary Of Contributions:**

The paper proposes AutoAnnotator, a two-layer automated annotation framework that combines LLMs (meta-controller) and SLMs (task specialists). The LLM layer handles model selection, automatic code generation, and secondary review of difficult samples, while the SLM layer conducts consensus-based annotation and iterative fine-tuning. Experiments on six sentiment and toxicity datasets demonstrate significant improvements in accuracy and cost efficiency.

Strengths:
1. Clear motivation: Tackles high costs and low domain-specific accuracy of LLM based annotation.The annotation is always expensive, which is an important topic.
2. Strong experimental results: Outperforms both standalone SLMs and LLMs across zero-shot, one-shot, CoT, and majority voting baselines.
3. Practical value: Demonstrates that lightweight models can anchor annotation while selectively leveraging LLMs.

Weakness:
1. The evaluation is restricted to sentiment and toxicity classification, which are relatively shallow NLP tasks. It remains unclear whether the framework generalizes to complex reasoning tasks (e.g., multi-label, causal inference) or non-text modalities (vision, speech).

**Audience:**

Yes

**Audience Explanation:**

Yes. I believe this approach (based on the results of extensive experiments) is suitable for the NLP task at this point, and annotation is always very expensive for humans. This work reduces the cost by introducing the framework.

**Claims And Evidence:**

Yes

**Claims Explanation:**

The experiments are conducted on several of the major LLMs, yielding consistent results.

**Requested Changes:**

I wanted to check if the work can be extended to non-text data or a multi-label dataset. But, I understand that may be a different topic.

---

> ### Author Response · Authors · 2025-11-05
> **Responding to the Reviewer TGDB‘s comments**
>
> We sincerely thank you for your positive recognition of the innovation and contributions of our work. Your affirmation is a great encouragement to us. We are also very grateful for your valuable feedback. We have carefully considered your comments and provided a detailed response below.
>
> **It remains unclear whether the framework generalizes to complex reasoning tasks (e.g., multi-label, causal inference) or non-text modalities (vision, speech).** Thank you very much for your valuable feedback. Applying our method directly to multi-label datasets or non-text data is indeed a different and more complex topic. We fully understand your core concern about the generalization of our approach, namely whether our framework is limited to relatively basic NLP tasks such as sentiment classification and toxicity classification, and cannot be extended to complex tasks.To address your concerns, we conduct experiments on more challenging mathematical tasks.  Specifically, we select two widely used mathematical reasoning datasets, MathQA and AQUA-RAT, as the annotation task, and three publicly available models, Mathstral-7B-v0.1, Qwen2.5-Math-7B-Instruct, and DeepSeek-math-7b-Instruct, as the SLMs. To validate the effectiveness of our method, we compare the performance of AutoAnnotator with that of individual SLMs, as well as with GPT-3.5-turbo with additional inference strategies such as Self-Consistency and DynaThink. As shown in the table below, AutoAnnotator significantly outperforms all baselines, further demonstrating its generalization ability.
>
> | Model | Acc on MathQA | Acc on AQUA-RAT |
> | :--- | :--- | :--- |
> | Mathstral-7B-v0.1 | 39.60% | 36.83% |
> | Qwen2.5-Math-7B-Instruct | 80.29% | 79.87% |
> | DeepSeek-math-7b-Instruct | 66.43% | 65.33% |
> | Self-Consistency (GPT-3.5-turbo) | 65.00% | 64.20% |
> | DynaThink+Self-Consistency (GPT-3.5-turbo) | 68.00% | 61.50% |
> | **AutoAnnotator** | **85.00%** | **83.39%** |

---

### Review · Reviewer_6sy8 · 2025-10-23

**Summary Of Contributions:**

The paper introduces AutoAnnotator, a fully automated annotation framework based on a novel multi-model collaborative annotation paradigm. The key contributions of the paper are to automatically choose k models among a large suite of SLM's (using a frontier LLM), then identify if a particular query is difficult and further finetune the SLM's in a continued fashion.

**Additional Comments:**

None

**Audience:**

No

**Audience Explanation:**

At this moment, the claims of this paper have glaring gaps.

**Claims And Evidence:**

No

**Claims Explanation:**

There are several glaring gaps in the paper

1. In most tasks, there will not exist a finetuned SLM in the first place which can cater to the task. Finetuning the SLM itself will require a large amount of data - where will that data come from if the task is quite niche?

2. The cost of 1656$ is not prohibitive depending on the usecase. However, often there are glaring issues in the quality of annotations from frontier LLMs itself espescially for niche yet important tasks such as NL (natural language) to SQL conversion in large databases. The authors have discussed about data annotation but what is the usecase of such data. If the purpose of such data is finetuning then the system generating the data can be directly used for inference. "High Quality" is a subjective term - however the data collected in this fashion will not be enough to improve the frontier/teacher LLM itself.

3. Experiments on more datasets is necessary - code generation and code verification in particular. The two tasks considered are very generic and does not cover the gamut of niche tasks for which data is a bottleneck in the first place.

4. The adaptive model selection layer seems to require 1.69 M LLM queries per task. Why is k fixed? Again, for a niche task, I would imagine that only 1 or 2 SLM's might exist if we are lucky.

5. How are you getting the data for the hard samples? There is a disagreement between the SLM's so they are not trustworthy and the LLM is worse than the SLM according to the paper? So where are the high quality responses for continual training coming from in the framework where there is no manual involvement?

**Requested Changes:**

Please provide detailed answers to my questions and also expand the coverage of experiments.

---

> ### Author Response · Authors · 2025-11-05
> **Responding to the Reviewer 6sy8‘s comments (1)**
>
> We are sincerely grateful for your thorough review and invaluable feedback. Your insights are instrumental to the refinement of our work. We have carefully considered every point you raised and have made corresponding additions and clarifications. Below, we offer a detailed, point-by-point response to each of your comments.
>
> **In most tasks, there will not exist a finetuned SLM in the first place which can cater to the task. Finetuning the SLM itself will require a large amount of data - where will that data come from if the task is quite niche?** Thank you for this very practical question. First, our framework primarily targets domains where specialized SLMs already exist. As discussed in our paper (see Figure 1(a)) and in the introduction, specially trained SLMs have already surpassed general-purpose LLMs on many specific tasks (such as sentiment classification and toxicity classification). The AutoAnnotator aims to leverage the high accuracy and low cost of SLMs while addressing their generalization limitations.
>
> Regarding the situation you mentioned, where a task is so niche that no pre-trained or fine-tuned SLM is available, in such a "cold start" scenario, a batch of labeled data is needed first. In this case, we can first use LLM or manually annotate to generate a batch of initial data for training an initial SLM. Then, the AutoAnnotator intervenes, continuously improving data quality and SLM performance through subsequent collaborative annotation and iterative optimization. It's worth emphasizing that for such niche tasks, some initial data is always required, whether through manual annotation or direct annotation using LLM. In contrast, using our framework can reduce some of the annotation costs. To demonstrate the feasibility of this strategy, we conduct an experiment. Specifically, we first manually label 2000 samples in the JPP-Sentiment dataset and use these models to fine-tune three SLMs. Then, we use the fine-tuned models to run the previous process. As shown in the table below, Our initial strategy, utilizing publicly available pre-trained SLMs, achieves an accuracy of 78.56%. The new strategy, by using an SLM fine-tuned only on a self-labeled dataset to model “niche tasks,” achieves an accuracy of 78.29%. This experiment demonstrates the feasibility and robustness of AutoAnnotator, even for "niche" tasks where pre-trained SLMs are not available.
>
> | Model          | Scene                          | Acc     |
> | :------------- | :----------------------------- | ------: |
> | GPT-3.5-turbo  | Downloaded from huggingface    | 78.56% |
> |       GPT-3.5-turbo         | Trained using self-labeled data| 78.29% |
>
> We add a new paragraph in Section 3.1 to discuss solutions when there are no pre-trained models for a specific task and demonstrate experimentally in Section 4.2.
>
> **The cost of 1656 is not prohibitive depending on the usecase.** You point out that 1,656 (for 100,000 examples) isn't prohibitively expensive. We agree that this depends on the use case. However, the strength of our framework lies in its scalability. For real-world projects requiring hundreds of millions of labeled examples, a 74.15% cost savings can transform an extremely expensive project (perhaps hundreds of thousands of dollars) into an economically viable one. Our focus is not on whether the absolute value of 1,656 is high, but on the relative cost savings that our framework can achieve.
>
> **Experiments on more datasets are necessary.** Thank you very much for your valuable feedback. Your suggestions for code generation and code verification are excellent examples. We study your suggestions in depth and find that annotating for these tasks is inherently difficult and rare. To address your core concern about the generalization of our method to complex reasoning tasks in this revision, we have replaced them with mathematical reasoning tasks. We believe that mathematical reasoning is a highly complex domain that can adequately test the effectiveness of our framework. Specifically, we select two widely used mathematical reasoning datasets, MathQA and AQUA-RAT, as the annotation task, and three publicly available models, Mathstral-7B-v0.1, Qwen2.5-Math-7B-Instruct, and DeepSeek-math-7b-Instruct, as the SLMs. To validate the effectiveness of our method, we compare the performance of AutoAnnotator with that of individual SLMs, as well as with GPT-3.5-turbo with additional inference strategies such as Self-Consistency and DynaThink. As shown in the table below, AutoAnnotator significantly outperforms all baselines, further demonstrating its generalization ability.
> | Model | Acc on MathQA | Acc on AQUA-RAT |
> | :--- | :--- | :--- |
> | Mathstral-7B-v0.1 | 39.60% | 36.83% |
> | Qwen2.5-Math-7B-Instruct | 80.29% | 79.87% |
> | DeepSeek-math-7b-Instruct | 66.43% | 65.33% |
> | Self-Consistency (GPT-3.5-turbo) | 65.00% | 64.20% |
> | DynaThink+Self-Consistency (GPT-3.5-turbo) | 68.00% | 61.50% |
> | **AutoAnnotator** | **85.00%** | **83.39%** |

---

> > ### Author Response · Authors · 2025-11-05
> > **Responding to the Reviewer 6sy8‘s comments (2)**
> >
> > **The data collected in this fashion will not be enough to improve the frontier/teacher LLM itself.** We want to clarify that the goal of our AutoAnnotator framework is not to improve LLM. The core of AutoAnnotator is to leverage the generalization ability of LLM as an expert guide to examine difficult samples that SLM cannot handle. We then use this high-quality labeled data to iteratively fine-tune SLM, aiming to improve its generalization ability and make it perform better in subsequent annotation work. Our final output is a high-quality labeled dataset.
> >
> >
> > **The adaptive model selection layer seems to require 1.69 M LLM queries per task.** Regarding your point that the adaptive model selection layer seems to require 1.69 M LLM queries per task, we would like to clarify: We use LLMs such as ChatGPT to recommend SLMs based on the Hugging Face platform. These LLMs already incorporate extensive knowledge about the Hugging Face ecosystem and model library in their training data. Therefore, when selecting models, the LLMs leverage their power knowledge base and reasoning capabilities to conceptually and intelligently filter and recommend Hugging Face's vast model space, rather than actually querying each of the 1.69M models individually. We have revised the above description in the revised manuscript to eliminate any potential misunderstandings.
> >
> > **Why is k fixed?** $k$ is not theoretically fixed, but is the default optimal value we determined through experiments. We perform ablation studies in Section 4.3 (Table 6). We test the cases using $k=2, 3, 4, 5$ on the JPP-Sentiment dataset. We find that the annotation performance is best when k = 3. Therefore, considering the computational cost and annotation accuracy, we use k = 3 as the default in this paper. Our framework is flexible and can still operate if the meta-controller layer finds only one or two relevant SLMs after searching. We have already clarified in the revised draft that k does not have to be equal to 3.
> >
> > **How are you getting the data for the hard samples?** As we mentioned in Sec3.2, when the consensus of multiple SLMs on a certain sample is below the threshold, we consider it a hard sample.
> >
> > **Where are the high quality responses for continual training coming from in the framework where there is no manual involvement?** Thank you for your insightful review. Your understanding that "LLM is worse than SLM" is based on Figure 1(a), which shows the average accuracy across all samples. On this average performance, SLM (as a domain expert) does indeed outperform LLM. However, the "secondary review" mechanism designed in our framework is precisely intended to separate out the difficult samples that SLM struggles with. Table 8 (Table 14 in the revised version) compares the annotation accuracy of SLM and LLM on the difficult sample pool. We found that LLM's annotation accuracy on the difficult sample pool is significantly higher than SLM's. This also explains why our framework can provide high-quality responses without human intervention.

---

### Decision · Action_Editor_uJuY · 2025-12-15

**Recommendation:** Accept as is

**Audience:**

Yes

**Audience Explanation:**

The problem of scaling automatic annotation is a crucial industrial concern that is central to many data pipelines. This has led to extensive ML research. This paper addresses this issue directly and proposes a practical, cost-effective solution. Researchers interested in annotation and data-centric ML will find this work interesting.

**Claims And Evidence:**

Yes

**Claims Explanation:**

The revised manuscript provides extensive experiments across a wide range of tasks, including mathematical reasoning, sentiment analysis, and toxicity. These experiments clearly demonstrate the superiority of the proposed method over large models and strong SLM baselines. The authors have addressed the reviewer's concerns regarding the generality of the proposed approach, baselines considered, as well as the positioning of the paper. Overall, the claims made by the authors are convincing and clear.